# Experimental and Theoretical Investigations on Diamond Wire Sawing for a NdFeB Magnet

**DOI:** 10.3390/ma15093034

**Published:** 2022-04-22

**Authors:** Jia Liu, Zhenyu Zhang, Shengzuo Wan, Bin Wu, Junyuan Feng, Tianyu Zhang, Chunchen Zhou

**Affiliations:** 1Beijing Spacecrafts Manufacturing Factory, China Academy of Space Technology, Beijing 100094, China; liujia07105@sina.com; 2Key Laboratory for Precision and Non-Traditional Machining Technology of Ministry of Education, Dalian University of Technology, Dalian 116024, China; wwwsus1997@163.com (S.W.); a5548966329@163.com (B.W.); 15235927975@mail.dlut.edu.cn (T.Z.); 3Yantai Research Institute and Graduate School of Harbin Engineering University, Yantai 264006, China; zhoucc@hrbeu.edu.cn

**Keywords:** NdFeB, diamond wire sawing, vibration, surface formation mechanism, periodic waviness

## Abstract

The normal processing of sintered NdFeB magnets, used in many applied fields, involves diamond wire sawing. Due to the fact of its relatively lower hardness and high brittleness, the surface roughness and periodic waviness of the sawed surface have become a serious problem, but the surface formation mechanism is still unknown. In this work, a diamond wire sawing experiment with a NdFeB magnet was conducted while both the cutting force and the diamond wire lateral displacement were monitored. The vibration, the lateral swing of the wire and the cutting force were thoroughly analyzed. After the experiment, the surface morphology was carefully inspected under both a white light interferometer and SEM. It was discovered that the lateral swing of the diamond wire was the main cause of the periodic waviness on the surface, the PV of which was positively proportional to the normal cutting force. The surface morphology and surface roughness along the saw mark revealed that the vibration impact of ploughing/rubbing grits can induce the NdFeB grain to loosen off and cause more brittle fractures when the feed rate was 0.05 mm/min under wet cutting.

## 1. Introduction

As the third generation of rare earth permanent magnet materials, sintered NdFeB has been widely used because of its excellent magnetic properties. In recent years, electric vehicles have become an irreversible trend, requiring a large amount of NdFeB for their motors. However, the normal processing of NdFeB magnets involves diamond wire sawing and grinding. After diamond wire sawing, both the surface roughness and waviness are not good enough for application. Therefore, grinding is required to further reduce the surface roughness and eliminate periodic saw marks, which takes a significant amount of time. However, the forming mechanism of the cutting mark is still unknown; therefore, it is difficult to weaken or even remove cutting marks during the diamond wire cutting process.

The periodic waviness mark on the sawed surface is closely related to the reciprocating motion of the diamond wire. Many researchers have study how processing parameters influence the waviness mark’s formation. Qiu et al. [1] proposed that the saw mark is caused by periodic changes in the wire bow’s shape during the cutting process. The result showed that the wire bow’s deflection angle on two sides were different and changed as the reciprocating wire motion did, which implies that the wire bow’s shape also changes periodically. Jia et al. [2] investigated how the feed speed influences the waviness mark. It was discovered that the amplitude of the waviness as well as the period of the waviness decreased as the feed speed decreased. Huang’s study found that the length of the waviness was positively proportional to the feed speed. Maeda et al. [3] found that higher grit density and higher wire speed can effectively lower the waviness amplitude. In Teomete’s study [4], the formation of periodic waviness was explained as the result of the oblique cutting force. However, as the grit distribution is totally random, it is hard to explain why the total force direction would change periodically as the wire direction changes. Therefore, the cause of sawed surface waviness is still unknown.

Vibration of the wire is considered a very important factor in diamond wire sawing. Tang et al. [5] used a laser vibration monitor to obtain the vibration during the diamond wire saw cutting process, and a vibration prediction model was established and verified. The result shows that a higher feed speed leads to higher cutting force, which induces a higher wire vibration amplitude. As a result, the surface roughness is also higher when the vibration amplitude increases. Huang et al. [6,7] found that higher wire speed, lower feed speed and higher wire tension can result in a much lower vibration amplitude. In his study, kerf loss was discovered to be higher when the vibration amplitude was higher. To analyze the vibration characteristics, Wang et al. [8] created a vibration model focusing on the prediction of the vibration frequency and amplitude. Andhare et al. [9] used a similar modeling method to construct a vibration model for the wire electric discharge machining process, and the result showed that the wire speed had little influence on the vibration frequency response. In the above studies, the vibration is considered a key parameter during the cutting process. However, how the vibration contributes to the waviness as well as the rough surface formation process is still unknown.

For NdFeB magnets, the hardness is not as high as hard brittle crystalline materials such as single crystal silicon [10,11,12] or SiC [13,14,15], which are often the focus of diamond wire saw studies. As a powder sintered material, NdFeB magnets are composed of compacted grains with a size of a few micrometers including the normal NdFeB phase and the rich Nd phase between grains. Since the rich Nd phase, which serves as the binding phase, is very soft and chemically active, the binding force is not high enough and the NdFeB grains can easily be pulled out of the surface during grinding or cutting [16,17]. It was also discovered that the rich Nd phase was always the first to be etched during erosion by acid or even salt solution, resulting in the loosening of the NdFeB grain [18,19,20]. As a result, fractures found in NdFeB magnets always happen at the grain boundary, causing severe surface damage and high surface roughness during machining [21,22]. Due to the flexibility of the diamond wire, high-frequency vibration will consistently occur during the sawing process. Therefore, how high-frequency vibration may influence the material removal process and the surface formation mechanism of NdFeB magnets is still unknown.

In view of the previous study, the contribution of vibration to waviness and surface roughness is still not clear, while the diamond wire saw process study for NdFeB magnet is rare. Moreover, the surface formation mechanism of NdFeB is still unknown, especially under the vibration impact during the diamond wire sawing process. In this work, we proposed a method of monitoring the vibration and force during diamond wire cutting process of NdFeB. A series of cutting experiments were conducted under different coolant conditions, feed speeds and wire speeds. The sawed surface roughness as well as waviness was closely observed under SEM and a white light interferometer. Finally, the rough surface formation process was then analyzed according to the vibration and cutting force under different cutting parameters to provide a better understanding of the material removal mechanism and surface formation mechanism for the diamond wire sawing of NdFeB magnets.

## 2. Materials and Methods

In this study, diamond wire saw experiments were conducted on an STX-203 diamond wire saw machine from Shenyang Kejing Auto-instrument Co., Ltd, Shenyang, China. The cutting force was recorded with a dynamometer provided by Shenzhen Lizhun Sensor ltd. The sample was mounted on the dynamometer while the dynamometer was fixed on the cutting platform. The normal cutting force and the thrusting cutting force was recorded during the cutting process. To obtain the lateral vibration of the wire during the cutting process. The laser displacement sensor was placed on the front and the laser travels horizontally. The diameter of the laser spot was only 200 μm, which is smaller than the wire diameter. A small height adjustment platform was placed under the sensor so that the laser spot’s position could be adjusted vertically to point at the diamond wire. In this way, the sensor could obtain a horizontal displacement signal of the diamond wire during the cutting process. The experimental setup is shown in Figure 1.

The cutting diamond wire had a diameter of 250 μm, with #300 diamond grits electroplated on the wire. A N48H NdFeB magnet, provided by Advanced Technology & Materials Co., Ltd., Beijing, China, was selected as the sample in this work, and it had a size of 30 × 30 × 10 mm. During the cutting process, an ST12 water-based cutting fluid was supplied to the sample.

To investigate how the vibration and force change during the cutting process of the NdFeB magnet, different feed speeds, cutting speeds and cutting fluid conditions were selected for the experiment. The detailed experimental parameters are listed in Table 1; 0.05, 0.1, 0.2 and 0.3 mm/min were selected as the feed speeds. To investigate how the feed speed influences the cutting process under different spindle speeds, the spindle speeds were selected to be 200 and 300 rpm and the cutting speed were 1.3 and 2 m/s, respectively. Although dry cutting can obviously lead to poor surface roughness and a high cutting force, it is important for analyzing how the effect of cutting fluid during wire cutting process. In our study, due to the very low cutting efficiency and wire breaking problem, dry cutting with a 200 rpm spindle speed was not included in this study. As a result, dry cutting was only conducted for a 300 rpm spindle speed, while cutting with special cutting fluid was conducted for both the 200 and 300 rpm spindle speeds.

After the experiment, the surface morphology of the cutting surface was observed under a Zygo Newview 9000 3D optical surface profiler, after which the microscope images were processed in the Zygo Mx software to obtain both the areal surface roughness (Sa) and the surface roughness along the cutting direction (Ra) for multiple randomly selected sections and the surface morphology images. Finally, the sawed surface was observed under an FEI QUANTA 450 SEM to further analyze the surface formation mechanism.

## 3. Results and Discussion

### 3.1. Cutting Force

It can be seen from Figure 2a that the cutting force under dry cutting conditions was much larger than that under sufficient cutting fluid. For example, when the feed speed was 0.3 mm/min during dry cutting, the normal cutting force was twice as large as that during wet cutting, while the thrusting cutting force was three times as large. Coolant normally has three functions during the wire sawing process, which are cooling, lubrication and transportation of the cutting chips. During dry cutting, the lubrication is poor, henceforth, the cutting chip is stuck inside the cutting zone, further enhancing the friction between the wire and the workpiece, leading to a much larger cutting force. In this process, due to the existence of a large quantity of cutting chips blocking the grit from interacting with the surface, the sawing efficiency is also lower. When the material is being sawed by a diamond wire, there is a dynamic balance between wire deflection and the cutting efficiency. When the feed speed increases while other parameters remain the same, the cutting efficiency is not enough when the wire bow is not changed. Therefore, the material that is not sufficiently removed pushes back the cutting wire, resulting in a larger bow deflection angle (*θ*) and a larger than normal cutting force of *F* = *T* × sin*θ*. When the normal force on the wire increases, the cutting efficiency increases, and a new balance is established. For the same reason, as the cutting chip blocks the wire from efficiently cutting the material, the normal cutting force is much larger during dry cutting.

As shown in Figure 2b, when the feed speed increased during wet cutting, the cutting force increased. It can also be seen that the normal cutting force for 0.1 mm/min feed speed was larger than that under 0.2 mm/min when the spindle speed was 200 rpm. But the reason for that is still unknown. It is also worth noticing that both the normal force and thrusting cutting force were almost the same for 0.05 and 0.1 mm/min during wet cutting. When the feed speed decreased from 0.1 to 0.05 mm/min, the material needed to be removed was halved, the wire deflection should be much smaller, and the cutting force should be much lower. Therefore, the wire deflection had already reached a very small value, and it could be considered that the cutting efficiency almost reached a maximum level so that not much material would accumulate during the process when the feed speed was equal to or less than 0.1 mm/min. As shown in Figure 3, the wire bow was almost flat during the wet cutting process, which proves that the cutting efficiency was enough.

As for the influence of the spindle speed, it is worth noticing that the normal force for 300 rpm was slightly larger than that under 200 rpm as shown in Figure 2c,d. However, the thrusting force was smaller when the spindle speed was 300 rpm. When the feed speed was the same, but the wire speed was larger, the cutting efficiency increased as the amount of grits passing through the surface per second increased. Therefore, the wire bow deflection became smaller and the normal force decreased, but the lower wire speed also means a lower uncut chip thickness per grit and lower cutting chip transportation efficiency. As a result, the thrusting force was higher for the 200 rpm group due to the friction between grits, cutting chips and the surface.

### 3.2. The Wire Vibration under Different Cutting Conditions

Figure 4a shows a typical lateral motion curve of the wire during the cutting process, which is from the dry-300–0.3 group. To compare the lateral motion of the wire, the overall standard deviation of the original wire motion data was first calculated. The frequency domain response of the wire vibration was also obtained as shown in Figure 4c, which is also a useful tool for analysis [5]. It can be seen that the wire vibrated in high frequency (443 Hz) and the wire bow was slightly deflected in the lateral direction, causing the data to swing in a much lower frequency (2 Hz) with a much bigger amplitude. To better analyze the data and to separate these two factors, we separated the data vibration and data swing by low-pass and high-pass filter, respectively, as shown in Figure 4b. It can be seen that the vibration data constantly fluctuated around 0, with its amplitude and frequency unchanged compared to the original wire motion data. In addition, the swinging data had zero fluctuation, but the distance between the highest point and the lowest point of the curve was the same as the original wire motion data. For comparison, the standard deviation of the high-frequency component and the peak-to-valley (PV) value for the swing component, which is calculated by subtracting the lowest point from the highest point of the curve, were obtained.

According to previous studies [14,23], the vibration of the wire saw is excited by the constantly changing total cutting force direction. When the feed speed increased from 0.05 to 0.2 mm/min, the vibration amplitude as well as the lateral wire swing kept increasing as shown in Figure 5. However, it can be seen that the vibration of 0.3 mm/min was almost the same as for the 0.05 mm/min group, even though the swing motion was over six times the swing in the latter one. When the feed speed increased but the tension remained the same, the wire bow deflection kept increasing as the balance between the removal rate and feed speed was not met. Therefore, the wire bow deflection was very large when the feed speed increased to 0.3 mm/min and, thus, the wire was tightly pressed on the workpiece surface, suppressing its vibration.

For the wet cutting group, as shown in Figure 6, the vibration of the 0.05 mm/min group was the largest. After that, the vibration amplitude as well as the wire swing amplitude kept increasing as the feed speed increased. From the previous section, it was found that the cutting force was much smaller for the wet cutting process. Therefore, even for the 0.3 mm/min group, the wire bow deflection was still not large enough for the vibration to be suppressed. Instead, the vibration kept increasing as the cutting force increased due to the higher feed speed.

From Figure 7b,c, the variation trend for vibration and swing as the feed speed increased are clearer. The vibration for wet cutting with a 0.05 mm/min feed speed was the largest, even when compared to the dry cutting groups. As there was a balance between cutting efficiency and wire deflection, when the feed speed was only 0.05 mm/min, the wire was almost not deflected as shown in Figure 3. Therefore, the wire can vibrate freely even when a small cutting force was acted on the wire. But for much larger feed speeds, a larger wire bow was formed and, therefore, the vibration was slightly suppressed as the wire was tightly pressed onto the workpiece. Therefore, considering the special case for both the dry-300–0.3 group and the wet-300–0.05 group, it can be seen that the cutting force was not the only factor that can influence the vibration amplitude. When the wire bow deflection is too small, the vibration amplitude will increase heavily. But when the wire bow deflection is too large, the vibration will be suppressed as found in the dry cutting with 0.3 mm/min.

Figure 8 shows how the vibration and the swing motion were formed during the sawing process. During the sawing process, the total force acted on the wire constantly changed as a large amount of randomly distributed grit passes through the workpiece surface. Figure 8a shows the total force direction contributed by numerous grits with different grit positions and grit protrusions in a particular moment. It can be seen that the total force direction always deviated from the vertical direction by a small angle, resulting in a small lateral force acting on the wire and, henceforth, the lateral wire movement. However, as the total force direction changed frequently, the lateral movement direction also changed rapidly, resulting in a high-frequency vibration. Figure 8c shows the vibration of the wire at different locations and how the vibration amplitude could be recorded by the laser displacement sensor. The periodic lateral swing was also the result of the lateral sawing force, but the period for the swing strictly followed the reciprocating motion of the cutting wire. Therefore, the periodic lateral swing had nothing to do with the high-frequency change of the cutting force direction. Instead, the sawing machine’s assemble precision, especially the guiding wheel installation precision, should be considered the cause for the periodic swing of the wire. When the wire is cutting in different directions, due to the installation error of the guiding wheel, the stable wire’s position can be very different. Therefore, during the cutting process, there is always a periodic lateral force acting on the wire that changes direction as the wire cutting direction changes, creating such periodic waviness on the sample surface.

### 3.3. Surface Morphology

As shown in Figure 9b, the sawed surface roughness (Sa) increased as the feed rate increased under both cutting conditions. However, as shown in the surface morphology image, the surface roughness Sa was contributed to not only by the frequently found brittle fracture pits but also the heavy waviness that was perpendicular to the saw mark direction. Here, to separate the contribution of two factors, PV for the vertical profile and the surface roughness along the saw mark direction (Ra) were obtained, as shown in Figure 9a. In Figure 9c, the waviness PV increased as the feed rate increased. But the surface roughness Ra was slightly higher when the feed rate was 0.05 mm/min than 0.1 mm/min under wet cutting conditions. After 0.1 mm/min, the surface roughness kept becoming worse as shown in Figure 9d.

The surface morphology of the sawed surface from the white light interferometer is shown in Figure 10. It can be seen that the period of the waviness morphology also increased when the feed speed increased. With rough measurement and calculation, the waviness periods for the feed rates 0.05, 0.1, 0.2 and 0.3 mm/min were 37, 75, 150 and 225 μm, respectively. Considering that the time period for all wire saw experiments was 45 s, the feed distance for each group perfectly matched the waviness period. Therefore, the periodic swing of the wire was the cause for the waviness on the sawed surface.

From Figure 9c, it can be seen that the PV for the periodic waviness also increased when the feed speed increased. As shown in Figure 8b, the periodic swing of the wire was caused by the lateral force during the cutting process, which was the result of the guiding wheel installation error. When the feed speed increased, the cutting efficiency increased not only in the vertical direction but also in the lateral direction as the total cutting force increased rapidly. As a result, for the same cutting time period, the lateral cut-in depth was also larger under a high feed rate. For the same reason, the PV of the waviness in dry cutting was also larger. In many applied fields, the large periodic waviness on the sawed surface is not acceptable. Therefore, to lower the grinding time for removing such periodic waviness, a lower feed rate as well as sufficient lubrication are necessary.

### 3.4. Surface Formation Mechanism

For different cutting fluid conditions, the surface morphology was obviously much better for the wet cutting group. As shown in Figure 11a,b, the sawing mark was more obvious on the dry cutting group. During the dry cutting process, the cutting force was much bigger due to the poor lubrication and the friction between the wire, the cutting chip and the workpiece. As a result, the vibration was also much higher due to the high cutting force and, henceforth, the fracture zone was much deeper for the dry cutting groups.

As for the influence of the feed speed, the surface was much rougher when the feed speed was larger. Even though the surface roughness Ra along the saw direction was slightly higher for the 0.05 group than for the 0.1 feed rate group, the surface morphology under SEM for these two groups was very different as shown in Figure 12. As shown in Figure 12, more cracks and fracture pits could be found under the microscope for the 0.05 group. However, the surface for the 0.1 mm/min group was flatter with more ductile removal marks. It can be seen from Figure 12c that NdFeB grains were loosened and started to fall off the surface, but for the 0.1 group, the fracture zone was smoother.

To explain why more fracture behavior, especially the falling off of the grain, can be found in the 0.05 mm/min group, the average removal depth per grit needs to be calculated. As shown in Figure 13, during diamond wire sawing of the NdFeB magnet, two different material removal processes can be found. Depending on the protrusion height of each grit, the interaction between grit and workpiece can be classified into rubbing, ploughing and cutting modes. In diamond wire sawing, the average removal depth per grit (*d*_g_) can be expressed as:$$d_{g}=\frac{V}{n_{g}}=fw/Cv$$
where *f* is the feed speed, *w* the sample width, *C* the effective grit density, *V* the removed material volume and *v* the cutting speed. In previous section, it was shown that the cutting force was almost the same for the wet-300–0.05 and wet-300–0.1 group, which implies that the material removal rate was already excessive when the feed rate was 0.1 mm/min. For the 0.05 mm/min group, the feed rate was half of the 0.1 mm/min group and so was the average removal depth per grit, *d*_g_. As a result, more grit passed through the surface in a rubbing or ploughing mode, which did not induce material removal but still contributed to the cutting force; by raising the cutting force for the 0.05 mm/min group to almost the same level as 0.1 mm/min, even half of the material volume was removed. However, the laser displacement sensor detected that the vibration for the 0.05 mm/min group was almost twice than that for the 0.1 mm/min group. Therefore, these rubbing/ploughing grits frequently hit the sawed surface, resulting in NdFeB grain loosening off. The fracture pits where cracks had already formed became even deeper and more fractured under the vibration impact.

The specific cutting force, which refers to the average force acting on a single grit to remove a certain depth of material from the material, can also be used to explain the surface formation mechanism [24,25]. In diamond wire sawing, the specific cutting force can be expressed as:$$F_{h}=\frac{F_{n}}{Cd_{g}}=Fnv/fw^{2}$$

The specific cutting force for each group is shown in Figure 14, and it can be seen that the specific cutting force first decreased rapidly as the feed speed increased and then slowly increased during wet cutting, showing that there is always an optimal value for the specific cutting force to be minimal. When the specific force is larger, it means that the cutting force acting on a grit to remove the same amount of material is higher. As shown in Figure 14a,b, when the specific cutting force was larger, the vibration was also larger. Figure 14c shows that the surface roughness also had the same variation trend as the vibration and the specific cutting force. In fact, the diamond wire sawing process was more like grinding instead of cutting, where many grits interacted with the workpiece successively and simultaneously. Therefore, the force required to remove material not only includes the thrusting force of the cutting grits but is also contributed to by ploughing/rubbing interactions and the friction between the workpiece, cutting chips and the diamond wire. As shown in a previous section, vibration is another important factor that contributes to the cutting force. As a result of all these factors, the specific force becomes an effective tool that can provide insight into the diamond wire sawing process even when the cutting is still progressing. To minimize the surface roughness, promote the cutting efficiency and reduce energy loss, the cutting fluid condition and feed rate as well as spindle speed should be optimized based on the on-site cutting force and vibration monitoring results.

From the SEM images and the vibration data, it can be seen that the material removal process during diamond wire sawing of NdFeB included two modes. One was the cutting process, where the diamond grit scratched the surface deep enough to induce chip formation. In this process, crack initiation and propagation will occur if the protrusion height of the grit is large enough, which is similar to all other hard and brittle materials. However, unlike other brittle materials, the energy required for grain pull out is much smaller and the grain boundary fracture is much easier for NdFeB magnet. As a result, the ploughing and rubbing grits, which usually have no contribution to the material removal process, also causes one or a few grains to fall off the surface under vibration impact. As a result, a much rougher surface will form if the vibration amplitude is large, like the situation found during wet cutting with a 0.05 mm/min feed speed. Therefore, when processing materials, such as NdFeB magnets, the vibration should be considered as an important factor if a better surface quality is required. As a result, 0.1 mm/min should be considered as the best feed rate as the cutting efficiency is twice that of the 0.05 mm/min group, while the surface roughness and the periodic waviness were similar.

## 4. Conclusions

In this study, a diamond wire sawing experiment of NdFeB magnet had been conducted while both the cutting force and the wire lateral displacement were monitored. Combining with the surface morphology obtained under the white light interferometer and SEM, the periodic waviness formation mechanism and sawed surface formation mechanism were analyzed. The conclusions are as followed:When the feed rate increased in both the dry and wet conditions, the cutting force kept increasing as the balance between material removal efficiency and bow deflection reformed. The thrusting force for a lower spindle speed was larger, as the transportation of chips was less effective, but the normal force was smaller;The lateral wire displacement can be filtered into two components: one was the wire lateral swing following the reciprocating movement of the wire, the other one was the high-frequency vibration. The vibration for 0.05 mm/min during wet cutting was the largest; other than that, the vibration amplitude increased as the feed speed increased;The periodic waviness found on the sawed surface had the same length period as the calculation result of the distance the wire travelled for one reciprocating period. The variation trend of the swing amplitude of the lateral displacement also matched the PV value for the surface waviness, showing that the wire lateral swing was the main cause of the periodic waviness;The surface roughness Sa increased as the feed rate increased, which was mainly contributed by the waviness. But the roughness Ra along the saw mark direction was almost the same for the 0.05 and 0.1 mm/min groups during wet cutting. From the SEM image, it was discovered that the higher vibration for the 0.05 mm/min group caused more grain to loosen off and a more fractured surface. This implies that the surface formation of NdFeB magnets during diamond wire sawing includes not only cutting but also grain falling off due to the vibration impact.

## Figures and Tables

**Figure 1 materials-15-03034-f001:**
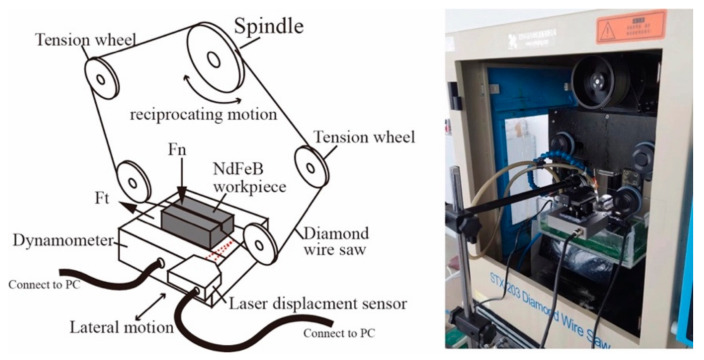
The experimental setup for all of the diamond wire sawing experiments.

**Figure 2 materials-15-03034-f002:**
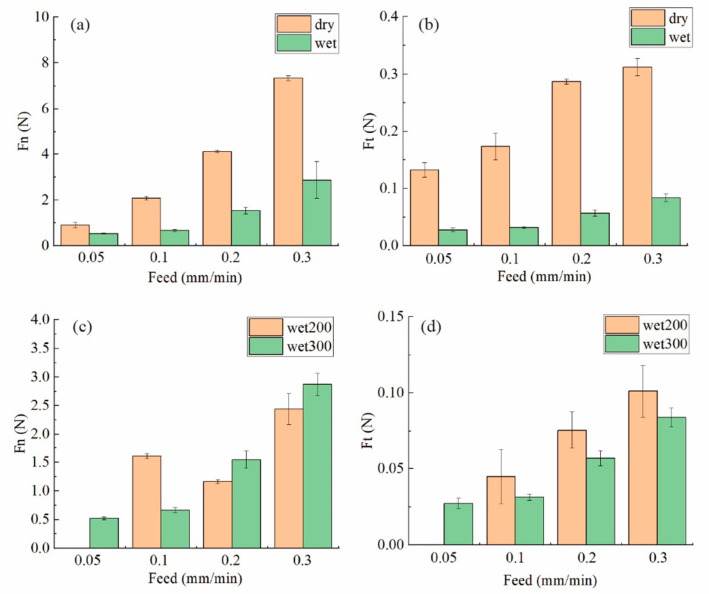
The thrusting force and normal force for (**a**,**b**) dry cutting and wet cutting conditions; (**c**,**d**) wet cutting with 200 and 300 rpm spindle speeds.

**Figure 3 materials-15-03034-f003:**
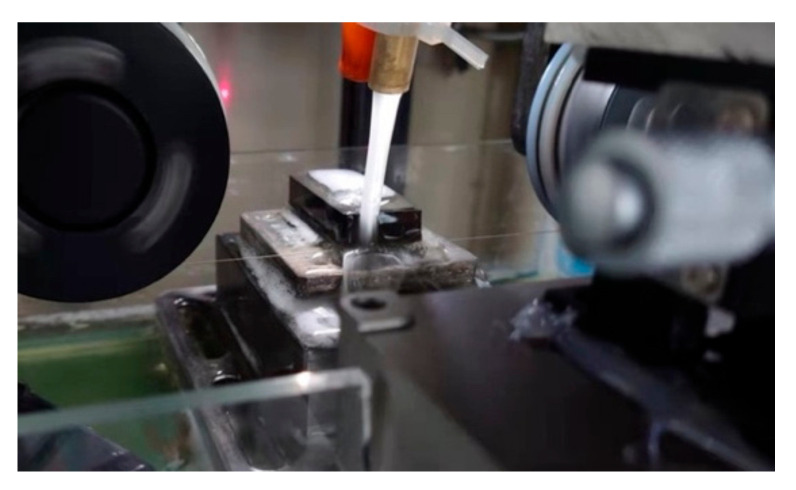
The flat wire bow during the 0.1 mm/min wire sawing process.

**Figure 4 materials-15-03034-f004:**
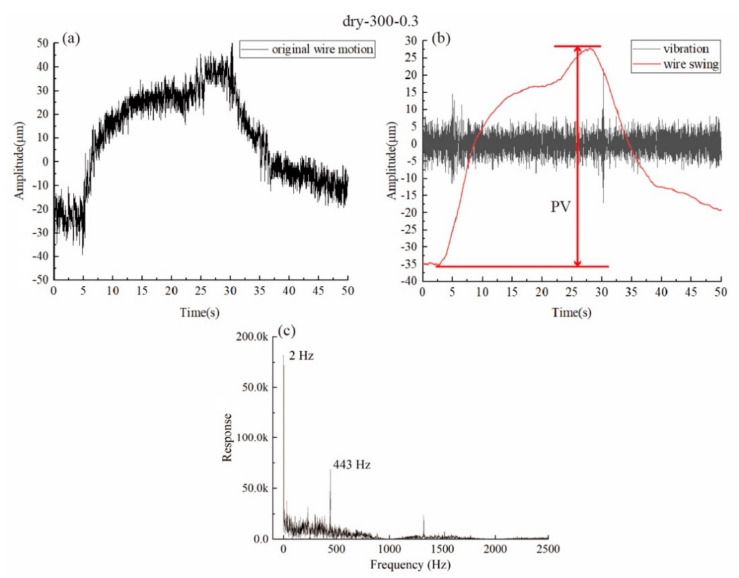
(**a**) The original wire motion; (**b**) the extracted vibration component and wire swing curve; (**c**) the frequency domain response for the original wire motion curve.

**Figure 5 materials-15-03034-f005:**
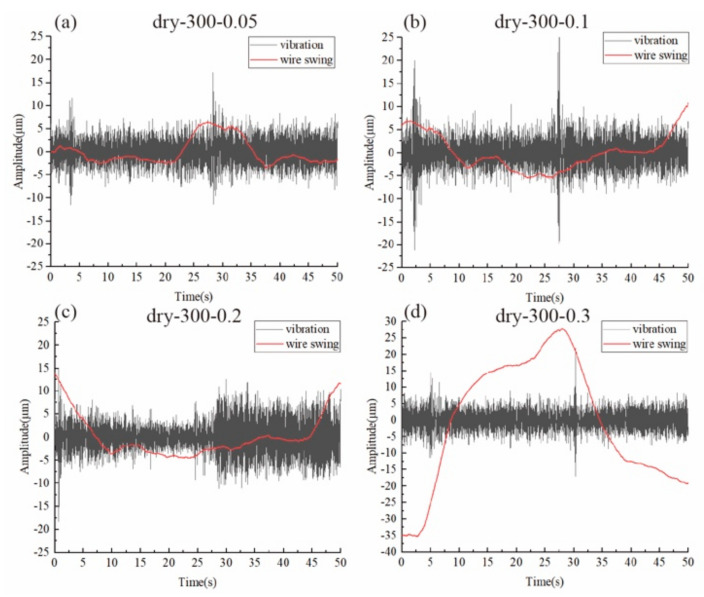
The wire vibration and wire swing motion during dry cutting.

**Figure 6 materials-15-03034-f006:**
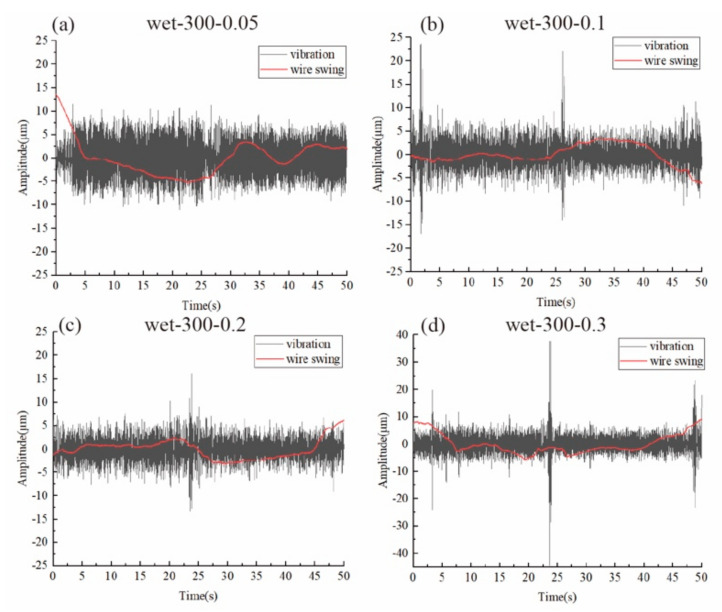
The wire vibration and wire swing motion during wet cutting.

**Figure 7 materials-15-03034-f007:**
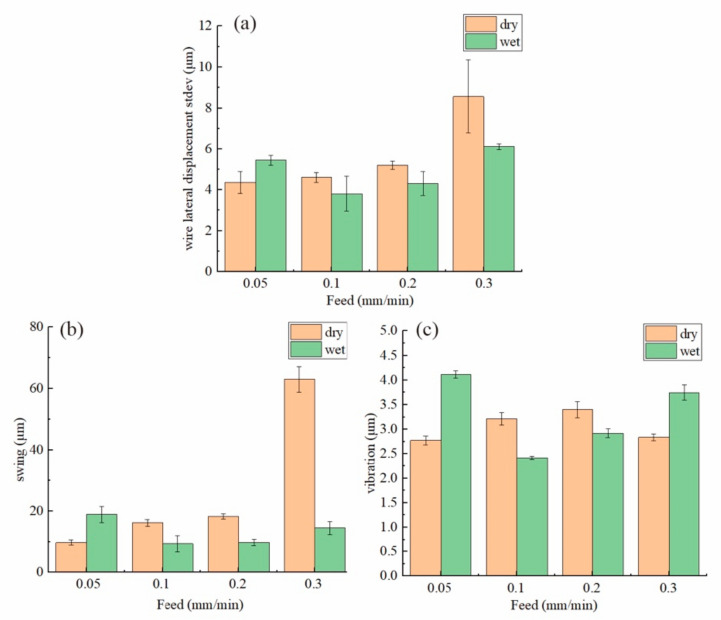
(**a**) The standard deviation of the wire motion data; (**b**) the PV for the swing component; (**c**) the standard deviation of the vibration component of the wire motion data.

**Figure 8 materials-15-03034-f008:**
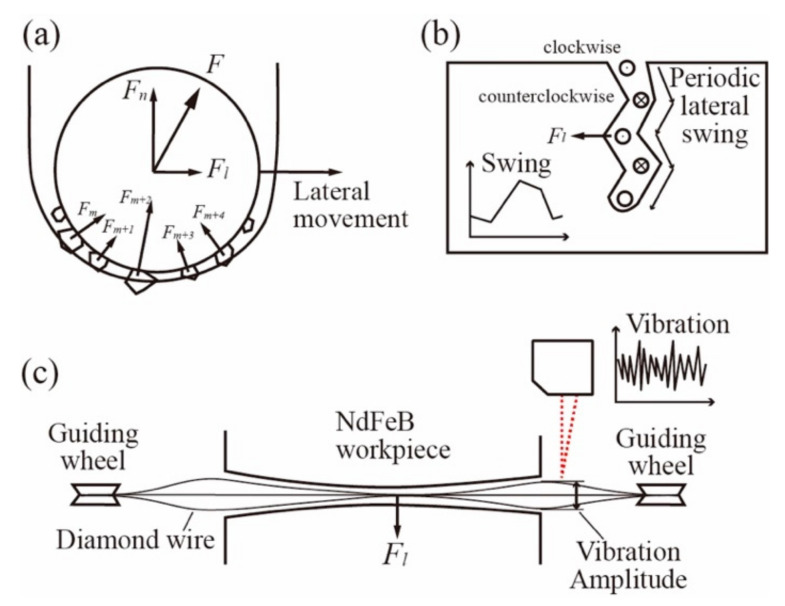
Illustrations of (**a**) the total force acted on the wire at a particular moment; (**b**) the formation of periodic lateral swing; (**c**) the test method and vibration amplitude at different location.

**Figure 9 materials-15-03034-f009:**
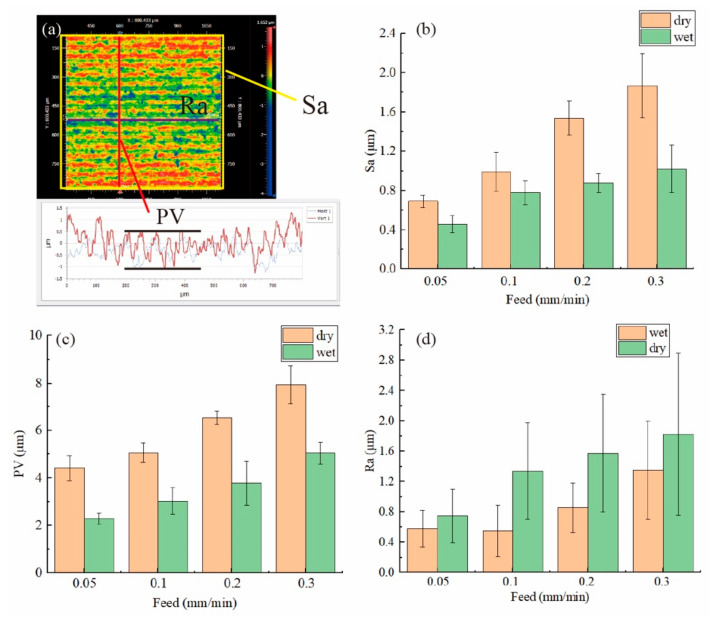
(**a**) A schematic of how all data were extracted from white light interferometer images; the surface roughness (**b**) Sa, (**c**) PV vertical to the saw mark and (**d**) Ra along the saw mark.

**Figure 10 materials-15-03034-f010:**
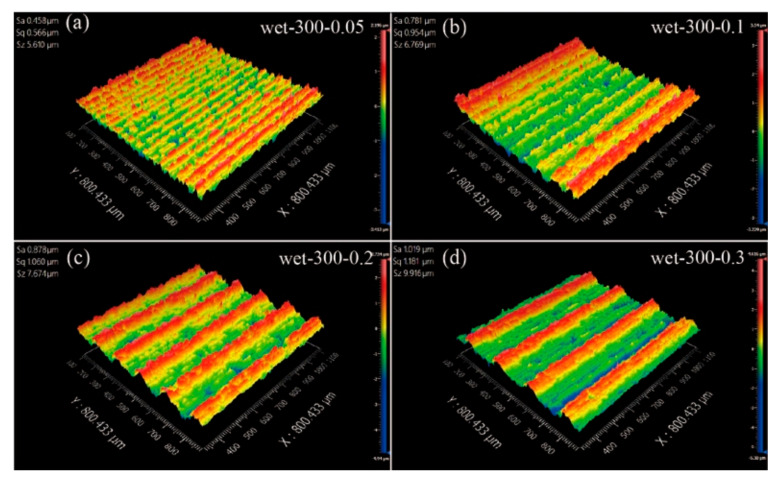
The surface morphology for wet cutting conditions with feed rates of (**a**) 0.05; (**b**) 0.1; (**c**) 0.2; (**d**) 0.3 mm/min.

**Figure 11 materials-15-03034-f011:**
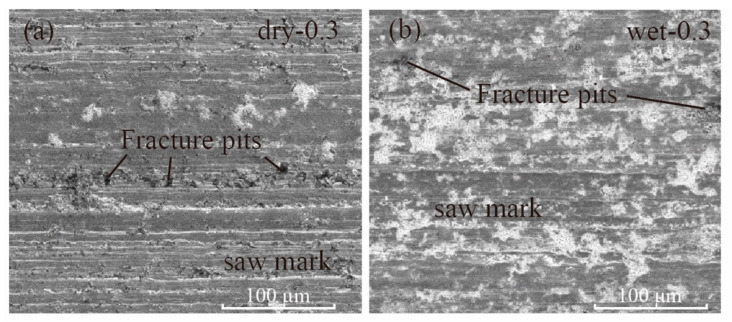
SEM images for (**a**) dry cutting and (**b**) wet cutting with a 0.3 mm/min feed speed.

**Figure 12 materials-15-03034-f012:**
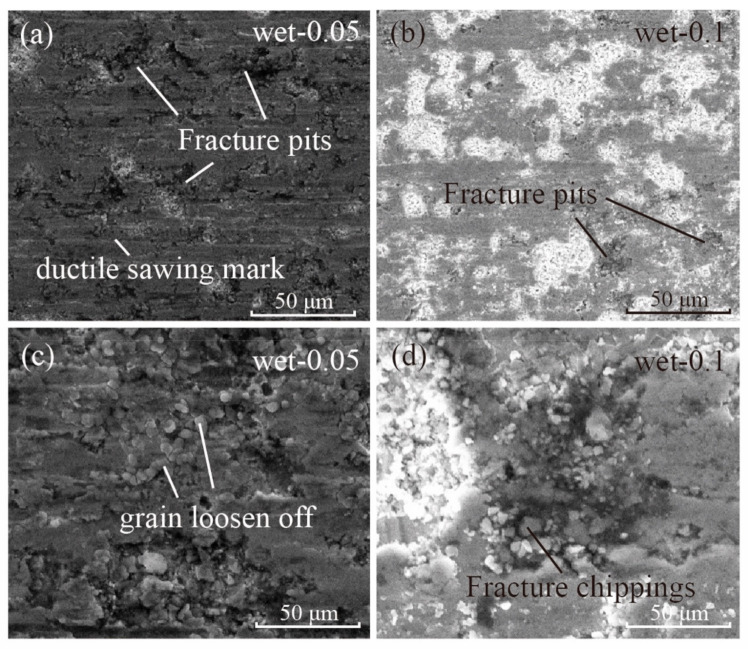
SEM images of the saw surface from the (**a**,**c**) 0.05 mm/min and (**b**,**d**) 0.1 mm/min groups.

**Figure 13 materials-15-03034-f013:**
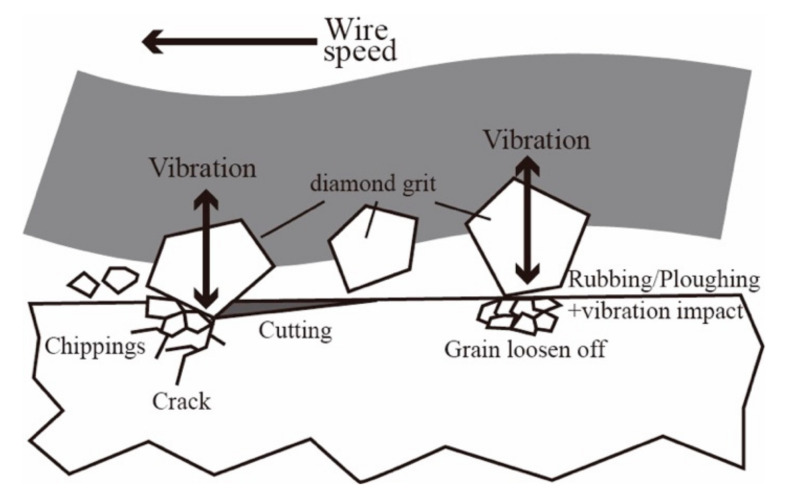
Two different mechanisms for the surface formation during diamond wire sawing of the NdFeB magnet considering the vibration.

**Figure 14 materials-15-03034-f014:**
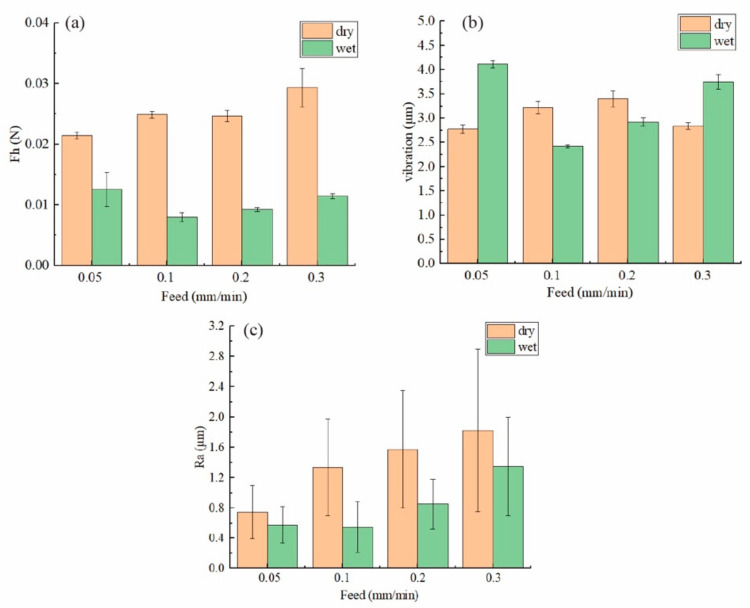
(**a**) The specific cutting force; (**b**) the vibration; (**c**) the surface roughness Ra along the saw mark for both the dry cutting and wet cutting conditions.

**Table 1 materials-15-03034-t001:** The cutting parameters.

Group Name	Coolant Condition	Spindle Speed (rpm)	Feed Rate (mm/min)
Dry-300–0.05	dry	300	0.05
Dry-300–0.1			0.1
Dry-300–0.2			0.2
Dry-300–0.3			0.3
Wet-200–0.1	wet	200	0.1
Wet-200–0.2			0.2
Wet-200–0.3			0.3
Wet-300–0.05	wet	300	0.05
Wet-300–0.1			0.1
Wet-300–0.2			0.2
Wet-300–0.3			0.3

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
