# Peer review of "Experimental and Theoretical Investigations on Diamond Wire Sawing for a NdFeB Magnet"

_materials, 2022, doi:10.3390/ma15093034_

Round 1

Reviewer 1 Report

The manuscript presents "Experimental and theoretical investigations on diamond wire sawing for an NdFeB magnet". 

Please find below my comments:

The literature review is very poor. An extended review is absolutely required. Additionally, there are only 18 positions of references. If the authors decide to publish in a high-quality journal, they should conduct a comprehensive review. It is the weak point of this manuscript.

The aim of the paper is not highlighted. 

It is nessecary to enter the designations of the devices used (country, city, etc.).

Author Response

Dear reviewer,

Thank you very much for your review comment. We have revied the manuscript according to your comments. Shown below is the detailed information on how I responded to the comments, highlighting the revisions made in the manuscript. When reading the information below, please note:

Text in italic style: the comments

Text in regular style: my responses to the comments.

Changes are highlighted in red color in the revised manuscript.

REVIEWER REPORT(S):

Comments to the Author

Recommendation: Publish after major revisions noted.

Comments: Please find below my comments:

  1. The literature review is very poor. An extended review is absolutely required. Additionally, there are only 18 positions of references. If the authors decide to publish in a high-quality journal, they should conduct a comprehensive review. It is the weak point of this manuscript.

We are deeply sorry for not providing a review that is closely related to the topic, which is the diamond wire sawing of the NdFeB magnet. However, it must be pointed out that there were zero previous research in this field. We did our best to look for any papers that are related to the subject, including cutting, milling, grinding or even polishing of the NdFeB magnet but failed, not to mention the diamond wire sawing process. As a result, we were only able to conduct a review based on the current diamond wire sawing process, including research on the wire vibration and surface formation mechanism.

To provide more information and better review, we have added a small paragraph to better address the differences between NdFeB and other hard brittle materials that were often sawed by diamond wire. In addition, we have also added a few more references on the subject of diamond wire saw process optimization.

  1. The aim of the paper is not highlighted.

Thank you for pointing that out. We have added a few lines in the last paragraph of the introduction section as well as the last section in the result and discussion section. The aim of this manuscript is to reveal the surface formation mechanism behind the wire sawing process of the NdFeB magnet. We have analyzed the vibration, the force and the wire lateral swing to show how the surface roughness and the periodic waviness was formed. In the last section, we have also reveal that the vibration impact can caused the grain to loosen off and induced more severe surface damage.

  1. It is nessecary to enter the designations of the devices used (country, city, etc.).

Thank you for your comment, we have added the device maker and model number for both the SEM and the white light interferometer. We have also added the software information for the image processing.

At last, we want to thank the reviewer for all the helpful comments, which has definitely improved the quality of this work.

Yours sincerely,

Zhenyu

Reviewer 2 Report

Referee report 
materials-1676154-peer-review-v1
Experimental and theoretical investigations on diamond wire sawing for an NdFeB magnet
Jia Liu, Zhenyu Zhang, Shengzuo Wan, Bin Wu, Junyuan Feng, Tianyu Zhang, Chunchen Zhou

This manuscript discusses the effect of diamond wire sawing on the surface of NdFeB-magnets. As cutting of such permanent magnets
is required to shape the magnets for many application, this is an important issue but often overlooked. The authors monitor the
force and the lateral movement of the wire, and study the microstructural details of the resulting sample surfaces.
Thus, the topic is well suited for Materials.
The present manuscript comprises 14 figures, 1 table, and 18 references are given.
The manuscript is well organized, well arranged and well written. The English is quite good; only minor changes are necessary.
All figures are well prepared, but the SEM and the interferometer images could be represented much larger to make the details
visible. As there is no page limit, this should be no problem.
Please remove all unnecessary information from the SEM images, and provide larger scale bars, which help the readers a lot.
The experimental section requires some more details about the optical microscopy, the white light interferometer and the SEM
like maker, model numbers, illumination, voltage, working distance, etc. 
Ref. [18] is not properly defined. Is this a book or a conference proceeding?

Overall, this manuscript contains interesting information. Thus, the manuscript may be accepted for publication provided that the
technical problems are treated well.

Author Response

Dear reviewer,

Thank you very much for your review comment. We have revied the manuscript according to your comments. Shown below is the detailed information on how I responded to the comments, highlighting the revisions made in the manuscript. When reading the information below, please note:

Text in italic style: the comments

Text in regular style: my responses to the comments.

Changes are highlighted in red color in the revised manuscript.

REVIEWER REPORT(S):

Comments to the Author

Recommendation: Publish after major revisions noted.

Comments: This manuscript discusses the effect of diamond wire sawing on the surface of NdFeB-magnets. As cutting of such permanent magnets is required to shape the magnets for many application, this is an important issue but often overlooked. The authors monitor the force and the lateral movement of the wire, and study the microstructural details of the resulting sample surfaces. Thus, the topic is well suited for Materials. The present manuscript comprises 14 figures, 1 table, and 18 references are given. The manuscript is well organized, well arranged and well written. The English is quite good; only minor changes are necessary.

  1. All figures are well prepared, but the SEM and the interferometer images could be represented much larger to make the details visible. As there is no page limit, this should be no problem.
    We have adjusted the figure size so that the details are more visible.
  2. Please remove all unnecessary information from the SEM images, and provide larger scale bars, which help the readers a lot.
    Thank you for your comment. We have deleted unnecessary information below the SEM and added larger scale bars.
  3. The experimental section requires some more details about the optical microscopy, the white light interferometer and the SEM like maker, model numbers, illumination, voltage, working distance, etc. 
    We have added the maker and the model for both the white light interferometer and the SEM. We have also added the software for image processing, which is also provided by the white light interferometer maker.
  4. Ref. [18] is not properly defined. Is this a book or a conference proceeding?
    Thank you for pointing that out, we have corrected the reference.

At last, we want to thank the reviewer for all the helpful comments, , which has definitely improved the quality of this work.

Yours sincerely,

Zhenyu

Reviewer 3 Report

The main idea behind the paper is interesting: the study of the surface formation mechanism in the case of diamond wire sawing of NdFeB magnet. The influence of the vibration, of the lateral swing of the wire and of the cutting force were analysed. The authors proposed a method for monitoring the vibration and force during the diamond wire cutting process of permanent magnet.

The title and the intentions declared in the abstract correspond to the contents of the paper. Some of the references could be more related to the subject of the paper.

Questions:

  1. Page 2, lines 82-83, the authors said: “To obtain the lateral vibration during the cutting process, a laser displacement sensor which has a 200 μm laser spot diameter was fixed to the machine chassis pointing”. Please explain in more detail the measurement of the lateral vibration using the laser displacement sensor;
  2. The Table 2 presents the design of experiments. I’m not satisfied with this DOE. It is not a fully factorial experiment, nor a half factorial experiment, nor a Taguchi DOE or other known DOE. Is it possible to evaluate the influence of different factors with this DOE?
  3. Page 3, the Materials and methods chapter: the entire paper is based on the surface morphology of the cutting surface. Please present in more detail (even also introduce a few photos) the microscope used in the measurement process and the SEM used in the evaluation of the surface formation mechanism;
  4. Page 3, lines 107-108, the authors said: “the cutting force under dry cutting condition was much larger than that under sufficient cutting fluid”. I think that this is obvious. In the cutting process without fluid, the friction forces will increase substantially.
  5. Page 4, Figure 2, a please correct the X axis title (Feed mm/mm). It is not visible in the figure.
  6. Page 5, Figure 4. The authors did not mention for which case they presented the wire motion, the vibration component and the frequency domain! Please be more explicit!
  7. Page 5, lines 165-167, authors said: “It can be seen from the standard deviation that the vibration was much larger when cutting under dry condition, and the vibration kept getting stronger when the feed speed increased under dry cutting condition”. Where can it be seen? They did not present any comparative results with the vibration under dry condition versus the vibration with fluid. The same remark for the feed speed. They only presented a chart (Figure 5) with synthetic results and some images (Figure 6). It is mandatory to present a comparison between the values obtained for both cases. Also, I consider that it is better to put figures 6 and 7 before figure 5.
  8. How can the authors explain the lack of the peak around 25 … 30 seconds in the Figure 6, b (for dry-300-0.2)? The same question for the case wet-300-0.05 (Figure 7, a);
  9. Please correct the titles of the figures and also of the chapter 3.3 (start with capitals).
  10. Can the authors explain the difference between saw roughness (Sa – arithmetical mean height) and surface roughness Ra?? The Sa is an extension of Ra. How can the authors explain that in Figure 9, Ra and Sa have a different behaviour??? Please also explain the PV abbreviation (maximum profile valley depth, I guess)!
  11. Page 9, lines 268-270, the authors said: “However, as shown in Figure 9 (d), the surface roughness Ra along the saw direction was slightly higher for the 0.05 group than the 0.1 feed rate group, but the surface morphology under SEM for these two groups were very different”. But Figure 9, b is a chart! I cannot appreciate the correctness of the statement based on a chart result.
  12. Page 12, lines 335-336, the first conclusion “The cutting force for dry cutting was much larger than for wet cutting due to the poor lubrication and transportation of chips” is obvious and can be eliminated.
  13. Which is the novelty of the present paper? Which are the differences between the cutting using a diamond wire sawing for an NdFeB magnet and other materials (magnetic or non-magnetic)?

I will provide the opportunity to the authors to extensively modify their manuscript.

Author Response

Dear reviewer,

Thank you very much for your review comment. We have revied the manuscript according to your comments. Shown below is the detailed information on how I responded to the comments, highlighting the revisions made in the manuscript. When reading the information below, please note:

Text in italic style: the comments

Text in regular style: my responses to the comments.

Changes are highlighted in red color in the revised manuscript.

REVIEWER REPORT(S):

Comments to the Author

Recommendation: Publish after major revisions noted.

Comments:The main idea behind the paper is interesting: the study of the surface formation mechanism in the case of diamond wire sawing of NdFeB magnet. The influence of the vibration, of the lateral swing of the wire and of the cutting force were analysed. The authors proposed a method for monitoring the vibration and force during the diamond wire cutting process of permanent magnet.

The title and the intentions declared in the abstract correspond to the contents of the paper. Some of the references could be more related to the subject of the paper.

Questions:

  1. Page 2, lines 82-83, the authors said: “To obtain the lateral vibration during the cutting process, a laser displacement sensor which has a 200 μm laser spot diameter was fixed to the machine chassis pointing”. Please explain in more detail the measurement of the lateral vibration using the laser displacement sensor;

Thank you for pointing that out, we have added some more description of the sensor setup.

To obtain the lateral vibration of the wire during the cutting process. The laser displacement sensor is placed on the front and the laser travels horizontally. The diameter of the laser spot is only 200 μm, which is smaller than the wire diameter. A small height adjustment platform was placed under the sensor so that the laser spot position can be adjusted vertically to point at the diamond wire. In this way, the sensor can obtain a horizontal displacement signal of the diamond wire during the cutting process.

  1. The Table 2 presents the design of experiments. I’m not satisfied with this DOE. It is not a fully factorial experiment, nor a half factorial experiment, nor a Taguchi DOE or other known DOE. Is it possible to evaluate the influence of different factors with this DOE?

We are sorry for the problem of the DOE. The initial thought of the experiment is to conduct a series of experiments to evaluate how the feed speed influence the cutting performance under different fluid condition, since we noticed that both the surface roughness and the waviness were very different under different feed speed during preliminary tests. To reveal the mechanism behind that, we measured both the vibration and the force of the cutting process. However, we considered that experiment involve only one factor was not enough to prove that the mechanism behind wire movement, material removal and surface formation is correct. Therefore, we have also included the wire speed and the cutting fluid condition to see if both of these factors might influence the cutting mechanism. During this process, we have encountered some problems. The wire kept breaking during cutting under dry condition with 200 rpm spindle speed. It was considered that the cutting efficiency was too low and therefore the wire bow was too large. At that moment, we considered that the comparison between wet-300 and dry-300 was sufficient. The large wire bow deflection and the small vibration in dry-300-0.3 group was also enough to show how large wire bow can suppress the wire vibration. We have also tried 250 rpm spindle speed but the machine was unstable and had large vibration. But as shown in the result and discussion section, spindle speed has no obvious influence on the mechanism. We were deeply sorry for not having a fully factorial or Taguchi DOE, but we believe that the experiment result and discussion we presented was enough for revealing the special mechiansm behind NdFeB magnet cutting, which was never reported in previous studies.

  1. Page 3, the Materials and methods chapter: the entire paper is based on the surface morphology of the cutting surface. Please present in more detail (even also introduce a few photos) the microscope used in the measurement process and the SEM used in the evaluation of the surface formation mechanism;

Thank you, we have included the detail information for the SEM and the white light interferometer microscope. We have also added the software for image processing and surface roughness calculation.

After the experiment, the surface morphology of the cutting surface was observed under a Zygo Newview 9000 3D optical surface profiler, after which the microscope images were processed in the Zygo Mx software to obtain both the areal surface roughness Sa, the surface roughness along the cutting direction Ra for multiple randomly selected sections and the surface morphology images. Finally, the sawed surface was observed under a FEI QUANTA 450 SEM to further analyze the surface formation mechanism.

  1. Page 3, lines 107-108, the authors said: “the cutting force under dry cutting condition was much larger than that under sufficient cutting fluid”. I think that this is obvious. In the cutting process without fluid, the friction forces will increase substantially.

Thank you for pointing that out. In normal machining processes like turning, milling or even grinding, both the normal force and the friction force are much larger without cutting fluid. It is the same case for the diamond wire cutting process, but the mechanism behind that was slightly different. During the wire cutting process, due to the flexibility of the diamond wire, the wire bow deflection increases as the cutting efficiency is less than the feed speed due to the insufficient chip transformation. When the wire bow deflection increases, the cutting force increases. We felt like it was the perfect moment to illustrate the mechanism behind wire bow deflection, cutting force and the balance between cutting efficiency and feed speed in the first paragraph of the discussion section. And the differences between dry cutting and wet cutting provided a good opportunity for that. In the following sections, we continue to reuse this mechanism to explain why the cutting force changes when other parameters varies, therefore, it was important for us to point out a phenomenon that is very obvious.

  1. Page 4, Figure 2, a please correct the X axis title (Feed mm/mm). It is not visible in the figure.

Thank you for your comment. We have adjusted the axis title.

  1. Page 5, Figure 4. The authors did not mention for which case they presented the wire motion, the vibration component and the frequency domain! Please be more explicit!

The wire lateral motion, the separated data and the frequency domain response were obtained from the dry-300-0.3 group, which was considered as a typical case to show how all these data were calculated and obtained.

  1. Page 5, lines 165-167, authors said: “It can be seen from the standard deviation that the vibration was much larger when cutting under dry condition, and the vibration kept getting stronger when the feed speed increased under dry cutting condition”. Where can it be seen? They did not present any comparative results with the vibration under dry condition versus the vibration with fluid. The same remark for the feed speed. They only presented a chart (Figure 5) with synthetic results and some images (Figure 6). It is mandatory to present a comparison between the values obtained for both cases. Also, I consider that it is better to put figures 6 and 7 before figure 5.

Thank you very much for pointing out that the discussion here is not properly arranged. We have already rearranged the whole 3.2 section here to better illustrate how the vibration and the periodic swinging data is obtained and how the synthetic result was obtained. Then, we have rewritten the statement comparing the vibration and swing motion.

Please refer to the revised manuscript for all the rewritten lines, which are marked in red.

  1. How can the authors explain the lack of the peak around 25 … 30 seconds in the Figure 6, b (for dry-300-0.2)? The same question for the case wet-300-0.05 (Figure 7, a);

Thank you for pointing that out. The vibration peak always happened during the wire direction change phase. However, the peak does not occur during every cycle as shown in the following figure. During this phase, the wire speed first decreases to 0 then increases to its maximum speed, the wire tension fluctuation may occur during this process due to the sudden increase of the thrusting force. But as the speed become stable, the cutting process and the wire vibration becomes more stable. Here, on our machine, the spindle speed was not too high and therefore the acceleration process was only 4~5 seconds, which was only a small fraction of the whole 46 s cutting cycle. Moreover, no obvious surface defect was found connected to such vibration peak, therefore it was not mentioned in this study.

  1. Please correct the titles of the figures and also of the chapter 3.3 (start with capitals).

We have corrected the mistakes and have check the whole manuscript for similar problems.

  1. Can the authors explain the difference between saw roughness (Sa – arithmetical mean height) and surface roughness Ra?? The Sa is an extension of Ra. How can the authors explain that in Figure 9, Ra and Sa have a different behaviour??? Please also explain the PV abbreviation (maximum profile valley depth, I guess)!

Surface roughness Ra is the roughness calculated by a linear profile, which can be expressed as:

The surface roughness Sa is the roughness of the whole area, which is calculated through:

If we consider the x direction is the wire cutting direction and y direction the feed direction, Sa is not only contributed by the surface roughness, but also by the waviness along the feed direction. However, as stated in the manuscript, we hope to separate the periodic waviness along the feed direction and the real surface roughness Ra along the cutting direction, because Ra is the one parameter that can reflect how rough the surface was under different material removal processes. We must point out that all these values, including Ra and Sa, were obtained by the Zygo image processing software. As shown in the figure, areal surface roughness Sa is read from the top-left corner, while horizontal Ra(along the cutting direction) can only be obtained when a linear sectioning is created by drawing a line.

As for the PV value, it stands for Peak-to-Valley value for the surface profile, which can be calculated by subtracting the highest point to the lowest point in one waviness period.

It can be seen that the vibration data constantly fluctuates around 0, with its amplitude and frequency unchanged when compared to the original wire motion data. Also, the swinging data have zero fluctuation but the distance between the highest point and the lowest point of the curve is the same as the original wire motion data. For comparison, the standard deviation of high frequency component and the peak-to-valley(PV) value for the swing component, which is calculated by subtracting the lowest point from the highest point of the curve, were obtained and compared for all groups as shown in Figure 7.

  1. Page 9, lines 268-270, the authors said: “However, as shown in Figure 9 (d), the surface roughness Ra along the saw direction was slightly higher for the 0.05 group than the 0.1 feed rate group, but the surface morphology under SEM for these two groups were very different”. But Figure 9, b is a chart! I cannot appreciate the correctness of the statement based on a chart result.

We are sorry for the misleading statement here. What we were trying to say is that Figure 9 shows the Ra value for two different groups, while Figure 12 shows the surface morphology under the SEM. We have corrected the statement here.

Even though the surface roughness Ra along the saw direction was slightly higher for the 0.05 group than the 0.1 feed rate group, the surface morphology under SEM for these two groups were very different as shown in Figure 12.

  1. Page 12, lines 335-336, the first conclusion “The cutting force for dry cutting was much larger than for wet cutting due to the poor lubrication and transportation of chips” is obvious and can be eliminated.

We have modified the first conclusion and deleted the first sentence.

  1. Which is the novelty of the present paper? Which are the differences between the cutting using a diamond wire sawing for an NdFeB magnet and other materials (magnetic or non-magnetic)?

Firstly, the diamond wire sawing of NdFeB magnet has not been reported in previous research, therefore the material removal mechanism as well as the surface formation mechanism is still unknown for such material. The separation of surface roughness along the cutting direction and the periodic waviness also helps explaining the influence from vibration and periodic wire bow swinging, which is also a new analyze method and new conclusion.

NdFeB magnet is a product of powder sintering, which has a low but not zero porous rate(10%~20%) and is consisted of normal NdFeB grain phase and rich Nd phase between grains. It is not a typical hard material and due to its powder sintering nature, the combining force between grains were very small. Therefore, when compared with normal hard brittle crystalline materials sawed by diamond wire saw, like single crystal silicon and SiC, NdFeB is very soft and brittle as the grain can be easily pulled out. As a result, the energy required for removing grains from the surface is much smaller than inducing a crack and supporting the crack propagation in other brittle materials. It is from the SEM images that we were able to confirm the grain loosen off and falling off phenomenon and verified the material removal mechanism and surface formation mechanism behind sawing of NdFeB. During diamond wire cutting of other hard brittle materials, the vibration was never considered a key factor during the surface formation, except for ultrasonic vibration. But for NdFeB, it was discovered that vibration contributes to the surface formation process as vibration impact can induce grain loosen off. Therefore, we consider that the contribution of this manuscript to be revealing the mechanism behind the cutting process of NdFeB, which was never reported in previous reseach.

From the SEM images and the vibration data, it can be seen that the material removal process during diamond wire sawing of NdFeB includes two modes. One is the cutting process, where the diamond grit scratches the surface deep enough to induce chip formation. In this process, crack initiation and propagation will occur if the protrusion height of the grit is large enough, which is similar to all other hard and brittle materials. However, unlike other brittle materials, the energy required for grain pull out is much smaller and the grain boundary fracture is much easier for NdFeB magnet. As a result, the ploughing and rubbing grits, which usually have no contribution to the material removal process, also causes one or a few grains falling off the surface under vibration impact. As a result, a much rougher surface will form if the vibration amplitude is large, like the situation found during wet cutting with 0.05 mm/min feed speed. Therefore, when processing materials like NdFeB magnet, the vibration should be considered as an important factor if a better surface quality is required. As a result, 0.1 mm/min should be considered as the best feed rate as the cutting efficiency is twice as the 0.05 mm/min group while the surface roughness and the periodic waviness were similar.

Thank you very much for asking such deep and meaningful question. It felt like we have not been able to have such discussion until now. Therefore, we have added a paragraph in the final section, devoted to addressing the differences between NdFeB and other hard brittle materials.

At last, we want to thank the reviewer for all the helpful comments, which has definitely improved the quality of this work.

Yours sincerely,

Zhenyu

Round 2

Reviewer 1 Report

The comments were taken into account. Thank you very much. The quality of manuscript increased. 

Reviewer 3 Report

Accepted as it.